# Skin Malignancy Classification Using Patients' Skin Images and Meta-data: Multimodal Fusion for Improving Fairness

**Ke Wang**[1]
**Ningyuan Shan**[1]
**Henry Gouk**[1]
**Iris Szu-Szu Ho**[1] *
[1] *School of Informatics, University of Edinburgh, United Kingdom*

**Editors:** Accepted for publication at MIDL 2024

## Abstract

Skin cancer image classification across skin tones is a challenging problem due to the fact that skin cancer can present differently on different skin tones. This study evaluates the performance of image only models and fusion models in skin malignancy classification. The fusion models we consider are able to take in additional patient data, such as an indicator of their skin tone, and merge this information with the features provided by the image-only model. Results from the experiment show that fusion models perform substantially better than image-only models. In particular, we find that a form of multiplicative fusion results in the best performing models. This finding suggests that skin tones add predictive value in skin malignancy prediction problems. We further demonstrate that feature fusion methods reduce, but do not entirely eliminate, the disparity in performance of the model on patients with different skin tones.

**Keywords:** Bias Reduction, Fairness Evaluation, Fusion Models, Malignancy Classification, Multimodal Learning, Patient Data Integration, Skin Cancer

## 1. Introduction

Skin cancer is one of the top five most common cancers in the United Kingdom (UK), with around 17,845 new cases in 2019 (World Cancer Research Fund, 2019). Of all of the skin cancer types, around 5% of skin cancers are melanoma skin cancer and the other 95% include the Basal Cell Carcinoma (BCC) and Squamous Cell Carcinoma (SCC) and others (Diepgen and Mahler, 2002). Early diagnosis of a skin cancer improves patient outcomes and is correlated with over 90% survival (Cancer Research UK, 2015). However, once disease progresses beyond the skin, survival is poor, falling to 32% when it metastasizes to distant organs (American Cancer Society, 2023). This highlights the importance of early skin cancer screenings and detection.

Most of the computer aided techniques for skin cancer diagnosis are focused on analyzing images of skin lesions (Xu et al., 1999; Jain et al., 2015). Convolutional neural network (CNN), for example, has been applied successfully to various tasks in skin cancer diagnosis using Dermoscopy or clinical images (Refianti et al., 2019; Zhang et al., 2020; Dorj et al.,

---

* Corresponding author: iris.s.s.ho@ed.ac.uk

2018; Mijwil, 2021). Other trained models that have been used for skin cancer image classification include Artificial Neural Networks (ANNs) (Jaleel et al., 2012; Kanimozhi and Murthi, 2016) and Long Short Term Memory (LSTM) (Mahum and Aladhadh, 2022; Yunandar and Irianto, 2022).

Unlike unimodal learning, multimodal learning is a relatively new field in skin cancer diagnostics. Multimodal learning involves the integration of multiple modalities, such as image data and medical records, to create more accurate predictions and diagnoses (Poria et al., 2017). Prior study has shown that multimodal fusion can facilitate fair and unbiased medical diagnostics by leveraging multiple sources of information (Zhou et al., 2021). Fusion is especially important in skin malignancy classification because skin cancer affects people of all skin types and colors, but certain groups, such as people with darker skin tones or manifestations of non-melanoma skin cancer, are often underdiagnosed or diagnosed later in the disease progression (Gloster Jr and Neal, 2006). To ensure fairness and improve model performance, it is critical to incorporate a diverse dataset that accurately reflects the population being studied and also with consideration for privacy concerns (Yan et al., 2023). Therefore, the research questions of this study aim to address are:

1. whether patient skin tone can add predictive value to a model that could, otherwise, only accommodate images?

2. which models are effective in predicting skin maglignancy equally between the three skin tone groups?

3. which models minimize the gap in performance across the three skin tone groups?

## 2. Related work

In the field of dermatological research, image-only models have been at the forefront of addressing biases associated with skin tones. a prior study by Correa-Medero et al. (2023) exemplified this approach by relying on image data, employing adversarial training to mitigate bias in skin color recognition—a method that significantly contributes to the ongoing efforts to refine bias mitigation in image-based skin cancer diagnosis models. Similarly, the works by Oguguo et al. (2023); Du et al. (2022) show the importance of visual data in fostering fairness, specifically targeting skin tone disparities. These efforts are complemented by Kinyanjui et al. (2020), who, while focusing on skin tone classification in the context of skin diseases, highlighted the critical role of skin color in dermatological studies.

Building upon these image-centric methodologies, the exploration of fusion techniques represents a promising avenue for enhancing model performance and fairness. To our knowledge, little attention has been paid to fuse patients' skin tones into a skin malignancy classification model. Unlike skin tones, age and sex were commonly used as input features, whereas previous research established that fusing age and sex into a model did not perform as good as an image-only model (Höhn et al., 2021). The existence of bias in skin cancer classification remains for people with different skin tones (Yuan et al., 2022). The use of fusion methods has been reported to be effective in addressing fairness in classification tasks (Zhou et al., 2021). In skin cancer research, late-fusion techniques are commonly adopted, which fuse the separately extracted data at the last stage of the network (Ge et al., 2017; Yap et al., 2018;

Kawahara et al., 2018; Singh et al., 2022). The late-fusion approaches tend to ignore the potential correlation in mixed feature space. To overcome the drawback of late fusion, early fusion can address some of the limitations of late fusion. In early fusion, the raw inputs from different modalities are combined to learn a joint representation of the different input modalities, which could improve the performance of a model by capturing complementary information from different sources (Heiliger et al., 2022; Vielzeuf et al., 2018). This can be exemplified by Chen et al. (2022) work where they proposed the MDFNet that used a combination of feature-level and decision-level fusion techniques to integrate clinical data into image data, and the fusion model achieved 80% accuracy (Chen et al., 2022). Building on prior work, the use of early fusion methods in this study demonstrates an approach to addressing fairness in classification models, and highlights the importance of considering the impact of different input features and modalities on the performance of the models.

## 3. Methodology

There were three models involved in this study: the Image-only Model (Figure 1A), the Feature Fusion Model (Figure 1B), and the Learning Feature Fusion Model (Figure 1C). These models were built on pre-trained architectures and utilized a common set of hyper-parameters during training to ensure a fair performance assessment (Table A2). In each of the three models, there are two main components: feature extraction and classification.

- Feature Extraction: This component involved extracting relevant features from the input data, which in this case are skin lesion images. In all three models, feature extraction was performed by a pre-trained model. The selection of pre-trained models are discussed in Model Training Process.

- Classification: Once the features were extracted, the next step was classification, where the models made predictions about skin malignancy. The sigmoid classifier learned patterns in the features extracted by the pre-trained model and predicted the class label (malignant vs benign) for each input.

### 3.1. Image Only Models

In our image-only model (Figure 1A), the preprocessed images were used as input for feature extraction using a pre-trained model, which was then converted into a one-dimensional vector for classification. The initial and subsequent linear layers served to reduce the dimensionality of the feature space, first to 1024, then to 512. This reduction in dimensionality helped improve computational efficiency and aided in capturing essential information while discarding potentially less relevant features. To capture complex data relationships, the architecture incorporated non-linearity by applying Rectified Linear Unit (ReLU) activation functions after each linear layer (Agarap, 2018). Additionally, to address overfitting, dropout layers with a dropout rate of 0.25 were added after each ReLU activation function. By randomly dropping out units, it helped prevent the model from relying too heavily on specific features or relationships, thus promoting more robust and generalized learning (Garbin et al., 2020). In the final stage of the architecture, the classifier consolidated the 512-dimensional feature space into a single output node. This simplified the classification

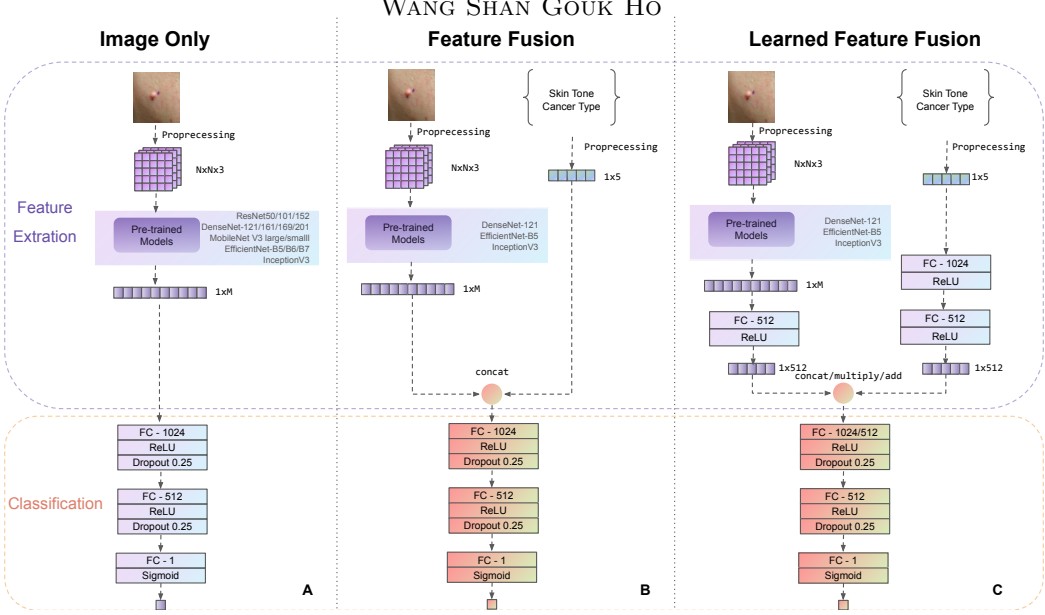

*The preprocessed image is represented as a three-dimensional tensor with dimensions NxNx3. The value of N depends on the specific pre-trained model and is determined based on the architecture and design choices of the model. The exact values of N can be found in Table A2. The extracted features are then transformed into a 1-dimensional vector of length M, with M being contingent on the inherent parameters of the applied pre-trained model.*

Figure 1: The architectures of the image-only, feature fusion, and learned feature fusion models

task to a binary decision: malignancy or non-malignancy. The output node represented the probability of the predicted malignancy classification.

### 3.2. Feature Fusion Models

Two fusion methods were used for this study: Feature fusion (Figure 1B) and Learned feature fusion (Figure 1C). In feature fusion, the features from image modality were extracted using pre-trained models. The image feature vector was combined with patients characteristics using concatenation. The combined features were then passed through the rest of the model for classification. In contrast, in learned feature fusion, a neural network was trained to extract features from image data and patients' metadata separately, and then combined them into a fused representation that captured the most important information from each source (Huang et al., 2020; Holste et al., 2021). One advantage of learned feature fusion is that it allowed the model to capture complex relationships between different sources of data (Huang et al., 2020). To understand how different fusion operations impacted predictive performance, we experimented with three fusion operations- concatenation, elemenwise addition and elementwise multiplication in the learned feature models, more details could be seen in Appendix B.

### 4. Experiments

### 4.1. Dataset and Baseline

Despite the importance of fairness in medical diagnostic, there remains a paucity of evidence on incorporating patient skin tones into malignancy classification. This motivated us to explore its potential impact. We used the Diverse Dermatology Images (DDI) dataset created by Daneshjou et al. (2022) (Daneshjou et al., 2022) for our experiments - a publicly

available and pathologically confirmed image dataset with diverse skin tones. More detailed informaiton about the dataset can be seen in Appendix C.

In the work by Daneshjou et al.(2022) (Daneshjou et al., 2022), they used the DDI dataset to train an image-only model with a pre-trained ResNet model [1] [2]. It's interesting to note that their model did not incorporate skin tone information yet achieved AUCs (Area Under the ROC Curve) ranging from 50% to 72%. The model by Daneshjou et al.(2022) (Daneshjou et al., 2022) was chosen as the baseline in our study, and our objective is to improve predictive performance and enhance fairness by considering different skin tones. For a comprehensive comparison of our results with other state-of-the-art methods using the DDI dataset, please refer to the summary table in the appendix (Table A1).

## 4.2. Preprocessing

The data were split into two sets: train and test. The train set consists of 590 images, while the test set contains 66 images. Ten-fold cross-validation was performed on the training set to evaluate the performance of the model. We used a 1:1:1 ratio of the three skin tone types to ensure that each skin tone type was represented equally in the test set (22 images per skin tone type). This can help to reduce the risk of bias and ensure fairness in the evaluation (Ricci Lara et al., 2022).

Before training, we resized all images to a consistent input size and normalized them using ImageNet's mean and standard deviation (Gouda et al., 2022; Pal and Sudeep, 2016). We applied data augmentation techniques, such as horizontal flip, random contrast adjustment, elastic transform, random scaling, and rotation to the cross-validation sets (Bloice et al., 2017). In the testing phase, we employed Test-Time Augmentation (TTA) with similar methods—horizontal flips, random contrast adjustments, and rotations—to mirror the training conditions. This approach not only ensured model evaluation consistency but also helped to improve adaptability to new, unseen image variations (Shanmugam et al., 2021; Gonzalo-Martín et al., 2021). As for the clinical data, the two categorical variables (skin tone and cancer type) were one hot encoded to convert them into numerical values that would be used for fusion.

## 4.3. Models Training Process

Since there were no pre-trained models trained specifically based on skin datasets, researchers often used popular pre-trained models to extract features from skin images. In this study, we experimented with a variety of pre-trained feature extractors, including Inception-V3 (Szegedy et al., 2016), ResNet-50/101/152 (He et al., 2016), DenseNet-121/161/169/201 (Huang et al., 2017), MobileNetV3-large/small (Howard et al., 2019), and EfficientNet-B5/B6/B7 (Melas-Kyriazi, 2023). Of image-only models, we trained the pre-trained models on the image data (1) without pre-trained weights, (2) with pre-trained weights, and (3) with frozen weights. Pre-trained models with frozen weights were found to perform better than those with pre-trained weights (unfrozen) and without pre-trained weights. Hence, in the following experiments, we focus on the models with frozen weights, particularly within

---

1. Github: https://github.com/DDI-Dataset/DDI-Code
2. Published paper: doi/epdf/10.1126/sciadv.abq6147

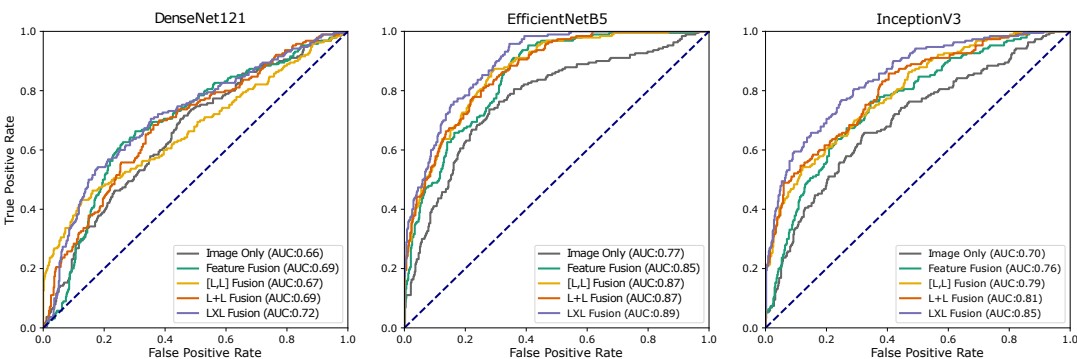

Figure 2: The ROC curve and AUC values for the DenseNet-121, EfficientNet-B5 and Inception-V3 models

feature extraction layers pre-trained on extensive datasets. This does not imply a complete freeze of all model weights, but a targeted approach to use pre-existing, generalized feature knowledge. Through the experiments, we found that Inception-V3, DenseNet-121 and EfficientNet-B5 performed better and thus those were used to extract rich image features for image classification tasks (Figure 2). These image features were then combined with patient characteristics and fed into three neural network architectures for further classification.

During model training, we used label smoothing and early stopping to address class imbalance and prevent overfitting (Lukasik et al., 2020). **Label smoothing** was used in the experiments to deal with imbalanced class (malignant: 171 cases vs benign: 485 cases). In label smoothing, instead of setting the target value for the correct class to 1 and all others to 0, a small positive value, usually around 0.1, was assigned to the negative class, and the target value for the true class was slightly reduced by the same amount. This encouraged the model to consider all classes equally and prevent overfitting to the majority class (Müller et al., 2019). **Early stopping** was used in the experiments to prevent overfitting and improve the generalization performance of the model(Prechelt, 2012), setting an upper training limit of 1000 epochs. However, the actual stopping varied, ranging from 40 to 100 epochs. For implementation details, see Table A2.

### 4.4. Evaluation Metrics and Fairness Assessment

To assess the overall discriminatory power of the models, the Area Under the Receiver Operating Characteristic (ROC) Curve, or AUC, was computed on a bootstrapped test set with 5000 iterations (Robin et al., 2011). The metric used to assess fairness of a model in this study is performance gap, which was calculated by taking the difference between the maximum and minimum AUC among the three skin tone datasets for each model (Ricci Lara et al., 2022). This metric quantifies the degree of bias and illuminates potential areas where model performance can be enhanced. In addition, the two aforesaid non-parametric statistical tests (the Wilcoxon and Friedman tests) were conducted to determine whether the differences in AUC values across skin tone groups were statistically significant or occurred by chance. By combining the measurement of performance gaps with a statistical test, it helped to gain insights into both the magnitude of the disparities and their statistical significance.

## 5. Results

### 5.1. Performance of image-only and fusion models

Of all image-only models, the models with EfficientNet-B5 and Inception-V3 had the best performance, with AUC 80.7% and 78.3%, respectively (Figure 2). The p-value of 0.119 was obtained for the model with EfficientNet-B5 in the Friedman test, and p-value of 0.202 obtained for the models with Inception-V3 and DenseNet-121. The image only model with the best performance (EfficientNet-B5) had the smallest performance gap between the three skin tones datasets (8.3%). This finding indicates that the EfficientNet-B5 image only model is less biased towards people with certain skin tones than Inception-V3 and DenseNet-121 models. By contrast, the largest performance gap (18.5%) was found in the model with Inception-V3.

In this study, fusion models (78.7%-92.0%) predicted skin malignancy better than image-only models (73.5%-80.7%) (Figure 2). This indicates that fusion models can potentially capture complex relationships between different features that may be missed by image only models alone. However, in the Wilcoxon test, we found that for DenseNet-121 there was no statistical difference between the image only model and fusion models (p-value > 0.05).

On the other hand, for EfficientNet-B5, learned feature fusion models significantly outperformed image only models (p-value < 0.05), whereas there was no difference between feature fusion and image only models. For Inception-V3, all fusion models significantly performed better than image only models except for the learned feature fusion model with concatenation operation.

Of the fusion models, it was found that learned feature fusion had better predictive performance compared to feature fusion models. This could be explained by the fact that features from the two modalities were used to learn a weight for each feature before combining into a weighted feature vector. This would allow the model to learn the optimal way to combine the features (Huang et al., 2020). Of all of the models, the best predictive performance was found in learned feature fusion model with EfficientNet-B5 (92.0%). Of the learned feature fusion models with three different fusion operations, the

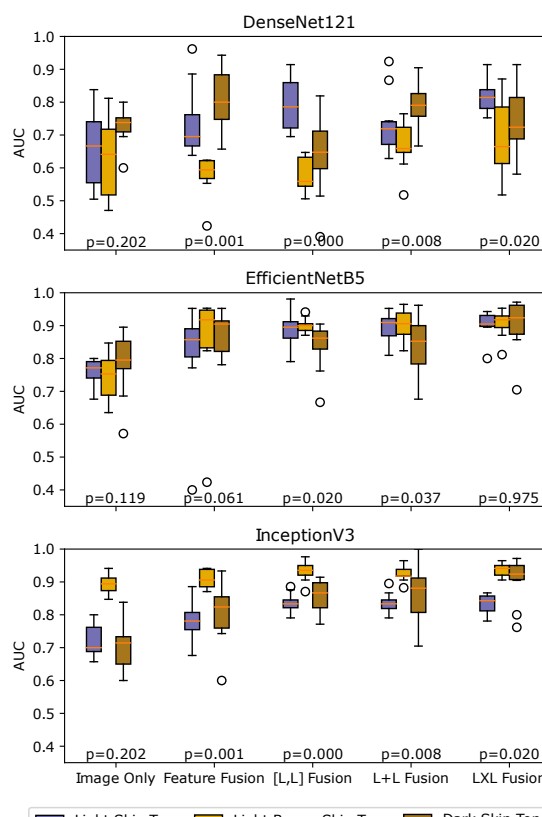

*p-value < 0.05 indicates the statistical difference in model performance between the three skin tones. See Table A5*

Figure 3: Comparing the performance of the fusion models with DenseNet-121, EfficientNet-B5 and Inception-V3 between three skin tones

models using elementwise multiplication fusion had better predictive performance than the models using concatenation and elementwise addition. Multiplication fusion operation is particularly useful when the modalities are expected to have a strong interaction or correlation (Liang et al., 2021; Fu et al., 2020; Liu et al., 2018). Hence, its effectiveness could imply that the image data and patients' meta-data were relevant and complementary.

### 5.2. Model performance in different skin tones

It was noted through experiments that models employing learned feature fusion, particularly multiplication fusion, generally exhibited a smaller gap compared to image-only models and feature fusion models (Figure 3). The smallest gap was observed in the EfficientNet-B5 model, with a disparity of merely 2.5%. By contrast, the models with DenseNet-121 and Inception-V3 did not yield similar predictive performance between the three skin tones (p-value < 0.05) (Figure 3). This reflects the fusion approach used was effective in combining information from the different subgroups without introducing any systematic bias or disproportionately weighting any particular subgroup.

## 6. Discussion and Conclusion

Our research used the unique DDI skin cancer database that includes skin tone information. We found that incorporating skin tone significantly enhanced model performance, showing skin color's vital role in predicting malignancies. While the fusion models improved performance, they did not eliminate disparities across skin tones, pointing to the need for further research for equitable skin cancer classification. In addition, employing "frozen weights" in our models led to quicker and more precise learning, particularly with a limited dataset. This is consistent with prior findings that fixed pre-trained weights are beneficial under data scarcity (Hinterstoisser et al., 2018). For fair model comparison, we used a uniform set of hyperparameters, offering clear insights into fusion strategies' impact on fairness and performance, though this uniformity might not represent each model's optimal settings.

In our study, we have laid the groundwork for the integration of skin tone information in skin malignancy prediction models. Future research should explore attention-based models with diverse, large datasets to balance performance and fairness, a challenge not pursued here. Implementing interpretability techniques, like feature importance analysis, could improve transparency and equity by revealing biases. Moreover, future experiments could expand the analysis to include various age groups (Rivers et al., 1989), gender diversity (Dao and Kazin, 2007), in addition to various skin tones, given our dataset's limited metadata, leading to a more comprehensive understanding of skin malignancies.

To summarize, this study represents an advancement in the field of skin malignancy prediction by being the first to combine skin color information with image data through a multimodal fusion approach. While the approach has proven effective in mitigating disparities and reducing bias in this study, future research is needed to delve deeper into the interpretability of the model (Kaur et al., 2020), and further explore the interaction between different data modalities.

## Acknowledgments

We would like to express our gratitude to the Centre for Doctoral Training in Biomedical Artificial Intelligence, which has provided invaluable support for our research endeavors. This programme, funded by the UK Research and Innovation (UKRI), has been instrumental in facilitating the advancement of our work. We are profoundly appreciative of the resources, guidance, and opportunities afforded to us through this initiative.

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

# Appendix A. Tables

Table A1: Comparison of Accuracy (%) between State-of-the-Art Methods and Our Model on the DDI Dataset

| Source | Model | Type of Model | All | Light skin | Light brown skin | Dark skin |
|--------|-------|---------------|-----|-----------|------------------|-----------|
| (Oguguo et al., 2023) | DeepDerm model | image only | 58.33 | 64.00 | 61.00 | 50.00 |
| | Pre-processing | image only | 60.33 | 66.00 | 59.00 | 56.00 |
| | In-processing | image only | 61.33 | 61.00 | 63.00 | 60.00 |
| | Post-processing | image only | 61.33 | 64.00 | 62.00 | 58.00 |
| (Du et al., 2022) | FairDisCo | image only | 83.33 | 83.78 | 84.31 | 81.82 |
| This work | EfficientNet-B5 | LXL Fusion | 84.54 [78.79, 89.39] | 84.09 [77.27, 90.91] | 84.09 [77.27, 90.91] | 85.45 [77.27, 90.91] |

*Oguguo et al. (2023) used the DeepDerm model, incorporating the Inception-V3 architecture and trained on the International Skin Imaging Collaboration (ISIC) dataset. Following the training, they employed three Bias Mitigation approaches to assess the model performance on the DDI dataset.*
*FairDisCo model was trained and tested on the DDI dataset using Disentanglement Contrastive Learning.*

Table A2: Parameter settings for each model

| Model | Input Size | Learning Rate | Batch Size | Optimizer | TTA | Pretrained Weights | Loss | Label Smoothing | Early Stopping |
|-------|-----------|---------------|-----------|-----------|-----|--------------------|------|-----------------|----------------|
| ResNet50 | 224x224x3 | 1e-4 | 8 | ADAM | 5 | ImageNet | BCEWithLogitsLoss | 0.1 | patience: 20 |
| ResNet101 | 224x224x3 | 1e-4 | 8 | ADAM | 5 | ImageNet | BCEWithLogitsLoss | 0.1 | patience: 20 |
| ResNet152 | 224x224x3 | 1e-4 | 8 | ADAM | 5 | ImageNet | BCEWithLogitsLoss | 0.1 | patience: 20 |
| DenseNet-121 | 224x224x3 | 1e-4 | 8 | ADAM | 5 | ImageNet | BCEWithLogitsLoss | 0.1 | patience: 20 |
| DenseNet-161 | 224x224x3 | 1e-4 | 8 | ADAM | 5 | ImageNet | BCEWithLogitsLoss | 0.1 | patience: 20 |
| DenseNet-169 | 224x224x3 | 1e-4 | 8 | ADAM | 5 | ImageNet | BCEWithLogitsLoss | 0.1 | patience: 20 |
| DenseNet-201 | 224x224x3 | 1e-4 | 8 | ADAM | 5 | ImageNet | BCEWithLogitsLoss | 0.1 | patience: 20 |
| MobileNetV3 large | 224x224x3 | 1e-4 | 8 | ADAM | 5 | ImageNet | BCEWithLogitsLoss | 0.1 | patience: 20 |
| MobileNetV3 small | 224x224x3 | 1e-4 | 8 | ADAM | 5 | ImageNet | BCEWithLogitsLoss | 0.1 | patience: 20 |
| EfficientNet-B5 | 456x456x3 | 1e-4 | 8 | ADAM | 5 | ImageNet | BCEWithLogitsLoss | 0.1 | patience: 20 |
| EfficientNet-B6 | 528x528x3 | 1e-4 | 8 | ADAM | 5 | ImageNet | BCEWithLogitsLoss | 0.1 | patience: 20 |
| EfficientNet-B7 | 600x600x3 | 1e-4 | 8 | ADAM | 5 | ImageNet | BCEWithLogitsLoss | 0.1 | patience: 20 |
| InceptionV3 | 299x299x3 | 1e-4 | 8 | ADAM | 5 | ImageNet | BCEWithLogitsLoss | 0.1 | patience: 20 |

Table A3: The AUC of image-only models with different pretrained weights

| Model | Without Pre-trained Weights | With Pre-trained Weights | With Frozen Weights |
|---|---|---|---|
| ResNet50 | 0.498 [0.349, 0.651] | 0.626 [0.473, 0.776] | 0.560 [0.404, 0.709] |
| ResNet101 | 0.517 [0.368, 0.669] | 0.653 [0.505, 0.796] | 0.628 [0.471, 0.772] |
| ResNet152 | 0.454 [0.297, 0.618] | 0.660 [0.505, 0.803] | 0.581 [0.418, 0.735] |
| DenseNet-121 | 0.538 [0.380, 0.692] | 0.758 [0.619, 0.879] | **0.735** [0.598, 0.858] |
| DenseNet-161 | 0.566 [0.408, 0.720] | 0.651 [0.512, 0.783] | 0.536 [0.393, 0.676] |
| DenseNet-169 | 0.592 [0.443, 0.742] | 0.691 [0.521, 0.844] | 0.538 [0.378, 0.700] |
| DenseNet-201 | 0.482 [0.329, 0.637] | 0.737 [0.595, 0.870] | 0.620 [0.482, 0.751] |
| MobileNetV3 large | 0.512 [0.358, 0.660] | 0.497 [0.343, 0.645] | 0.634 [0.487, 0.772] |
| MobileNetV3 small | 0.549 [0.389, 0.708] | 0.629 [0.467, 0.785] | 0.604 [0.462, 0.747] |
| EfficientNet-B5 | 0.566 [0.417, 0.709] | 0.775 [0.653, 0.880] | **0.807** [0.666, 0.926] |
| EfficientNet-B6 | 0.641 [0.493, 0.777] | 0.796 [0.664, 0.910] | **0.766** [0.630, 0.888] |
| EfficientNet-B7 | 0.540 [0.396, 0.681] | 0.805 [0.675, 0.916] | **0.812** [0.693, 0.914] |
| InceptionV3 | 0.429 [0.277, 0.590] | 0.757 [0.636, 0.867] | **0.783** [0.651, 0.897] |

Table A4: The AUC of image-only models with frozen weights between three different skin tones

| Model | Light skin | Light brown skin | Dark skin | Gap |
|---|---|---|---|---|
| ResNet50 | 0.648 [0.371, 0.876] | 0.529 [0.235, 0.812] | 0.562 [0.295, 0.819] | 0.119 |
| ResNet101 | 0.610 [0.352, 0.838] | 0.624 [0.235, 1.000] | 0.610 [0.305, 0.876] | 0.014 |
| ResNet152 | 0.667 [0.381, 0.905] | 0.576 [0.282, 0.847] | 0.448 [0.200, 0.714] | 0.219 |
| DenseNet-121 | 0.781 [0.562, 0.953] | 0.682 [0.412, 0.918] | 0.733 [0.467, 0.952] | 0.099 |
| DenseNet-161 | 0.419 [0.181, 0.676] | 0.565 [0.282, 0.835] | 0.676 [0.438, 0.895] | 0.257 |
| DenseNet-169 | 0.524 [0.238, 0.800] | 0.635 [0.353, 0.882] | 0.533 [0.229, 0.829] | 0.111 |
| DenseNet-201 | 0.619 [0.352, 0.857] | 0.506 [0.259, 0.753] | 0.752 [0.533,0.933] | 0.246 |
| MobileNetV3 large | 0.571 [0.286, 0.829] | 0.835 [0.647, 1.000] | 0.571 [0.286, 0.829] | 0.264 |
| MobileNetV3 small | 0.676 [0.438, 0.886] | 0.565 [0.247, 0.871] | 0.438 [0.171, 0.724] | 0.238 |
| EfficientNet-B5 | 0.810 [0.543, 1.000] | 0.765 [0.435, 1.000] | 0.848 [0.657, 0.981] | 0.083 |
| EfficientNet-B6 | 0.724 [0.457, 0.943] | 0.694 [0.388, 0.953] | 0.848 [0.648, 0.981] | 0.154 |
| EfficientNet-B7 | 0.714 [0.448, 0.933] | 0.729 [0.471, 0.953] | 0.971 [0.886, 1.000] | 0.257 |
| InceptionV3 | 0.733 [0.448, 0.962] | 0.918 [0.776, 1.000] | 0.762 [0.533, 0.943] | 0.185 |

Table A5: The AUC of fusion models with frozen weights

| Model | Fusion Methods | All skin tones | Light skin | Light brown | Dark skin | Gap |
|---|---|---|---|---|---|---|
| DenseNet-121 | Feature Fusion | 0.787 [0.652, 0.903] | 0.819 [0.600, 0.971] | 0.682 [0.329, 0.953] | 0.895 [0.733, 1.000] | 0.213 |
| | [L,L] Fusion | 0.675 [0.522, 0.819] | 0.829 [0.629, 0.971] | 0.482 [0.141, 0.847] | 0.676 [0.381, 0.924] | 0.347 |
| | LXL Fusion | 0.786 [0.662, 0.891] | 0.895 [0.724, 1.000] | 0.671 [0.329, 0.941] | 0.848 [0.638, 0.990] | 0.224 |
| | L+L Fusion | 0.719 [0.580, 0.838] | 0.733 [0.505, 0.924] | 0.682 [0.341, 0.953] | 0.819 [0.581, 0.990] | 0.137 |
| EfficientNet-B5 | Feature Fusion | 0.905 [0.828, 0.964] | 0.905 [0.762, 1.000] | 0.953 [0.824, 1.000] | 0.886 [0.714, 1.000] | 0.067 |
| | [L,L] fusion | 0.899 [0.820, 0.961] | 0.905 [0.762, 1.000] | 0.918 [0.765, 1.000] | 0.876 [0.695, 0.990] | 0.042 |
| | LXL Fusion | 0.920 [0.850, 0.974] | 0.943 [0.829, 1.000] | 0.918 [0.765, 1.000] | 0.933 [0.810, 1.000] | 0.025 |
| | L+L Fusion | 0.897 [0.816, 0.963] | 0.905 [0.762, 1.000] | 0.941 [0.824, 1.000] | 0.876 [0.705, 0.990] | 0.065 |
| EfficientNet-B6 | Feature Fusion | 0.875 [0.786, 0.952] | 0.838 [0.648, 0.981] | 0.871 [0.694, 0.988] | 0.952 [0.838, 1.000] | 0.114 |
| | [L,L] Fusion | 0.896 [0.812, 0.963] | 0.886 [0.714, 1.000] | 0.906 [0.753, 1.000] | 0.943 [0.819, 1.000] | 0.057 |
| | LXL Fusion | 0.903 [0.824, 0.966] | 0.886 [0.714, 1.000] | 0.894 [0.729, 1.000] | 0.952 [0.838, 1.000] | 0.066 |
| | L+L Fusion | 0.895 [0.809, 0.966] | 0.895 [0.733, 1.000] | 0.894 [0.741, 1.000] | 0.952 [0.838, 1.000] | 0.058 |
| EfficientNet-B7 | Feature Fusion | 0.880 [0.794, 0.953] | 0.829 [0.638, 0.971] | 0.800 [0.565, 0.988] | 0.990 [0.943, 1.000] | 0.190 |
| | [L,L] Fusion | 0.906 [0.826, 0.968] | 0.876 [0.705, 0.990] | 0.859 [0.624, 1.000] | 0.971 [0.886, 1.000] | 0.112 |
| | LXL Fusion | 0.905 [0.828, 0.970] | 0.867 [0.695, 0.981] | 0.847 [0.612, 1.000] | 0.962 [0.857, 1.000] | 0.115 |
| | L+L Fusion | 0.906 [0.830, 0.969] | 0.876 [0.714, 0.981] | 0.859 [0.624, 1.000] | 0.971 [0.886, 1.000] | 0.112 |
| InceptionV3 | Feature Fusion | 0.835 [0.731, 0.927] | 0.857 [0.676, 0.981] | 0.918 [0.776, 1.000] | 0.857 [0.667, 0.981] | 0.061 |
| | [L,L] Fusion | 0.837 [0.721, 0.934] | 0.819 [0.610, 0.971] | 0.918 [0.765, 1.000] | 0.924 [0.790, 1.000] | 0.105 |
| | LXL Fusion | 0.897 [0.811, 0.966] | 0.838 [0.638, 0.981] | 0.941 [0.812, 1.000] | 0.962 [0.876, 1.000] | 0.124 |
| | L+L Fusion | 0.839 [0.724, 0.933] | 0.857 [0.676, 0.981] | 0.929 [0.788, 1.000] | 0.886 [0.714, 1.000] | 0.072 |

[L,L]: Concatenation; [L X L]: Elementwise multiplication; [L + L]: Elementwise addition

## Appendix B. Fusion operations

We experimented three fusion operations: concatenation, elemenwise addition and elementwise multiplication in the learned feature models.

- Concatenation (L, L Fusion) involves combining the feature vectors from each modality into a single vector by appending them together (Liu et al., 2018; Liang et al., 2021). This method preserves all the information from each modality, but can lead to high-dimensional feature vectors that may be difficult to handle (Liu et al., 2018).

- Elementwise add (L + L Fusion) involves adding the corresponding elements of the feature vectors from each modality to create a new feature vector (Fu et al., 2020; Liu et al., 2018). This method is useful when the modalities capture complementary information but may not be useful when the modalities have conflicting informaiton.

- Elementwise multiply (L × L Fusion) involves multiplying the corresponding elements of the feature vectors from each modality to create a new feature vector (Liang et al., 2021; Fu et al., 2020; Liu et al., 2018). This method is useful when the modalities are expected to have a strong interaction or correlation but may not be useful when modalities are unrelated (Liu et al., 2018).

## Appendix C. Dataset

The DDI dataset contains 656 images and patient characteristics (skin tones, malignancy (target variable), and type of skin cancers). Malignancy is the binary target variable (malignant: 171 cases vs benign: 485 cases). As discussed earlier in the introduction, skin tones and type of skin cancers could potentially confound the classification of malignancy, and hence these patient features were included in our analyses. The skin tone variable

has three categories (light: 208 cases, light brown: 241 cases and dark: 207 cases). The dataset features two skin cancer categories: melanoma (21 cases) and non-melanoma (635 cases), across three skin tones. Light skin reports 7 melanoma and 201 non-melanoma cases, light brown skin has 7 melanoma and 234 non-melanoma cases, and dark skin presents 7 melanoma and 200 non-melanoma cases.

Clinical images and metadata were extracted from medical records. Skin lesions were taken using clinically-issued cameras. In the dataset, each patient has one image along with metadata (skin tones, clinical diagnosis, malignancy, and type of skin cancers). Malignancy is the binary target variable (malignant: 171 cases vs benign: 485 cases). Clinical diagnosis was excluded from the model training due to its strong association with the target variable 'malignancy'. As discussed earlier in the introduction, skin tones and type of skin cancers could potentially confound the classification of malignancy, and hence these patient features were included in our analyses. The skin tone variable has three categories (light: 208 cases, light brown: 241 cases and dark: 207 cases). Type of skin cancer variable consists of two categories: melanoma and non-melanoma.

