# OpenReview forum: "Skin Malignancy Classification Using Patients’ Skin Images and Meta-data: Multimodal Fusion for Improving Fairness"
_MIDL.io/2024/Conference — MIDL 2024 Poster_

### Official Review · Reviewer_NQzN · 2024-02-27

**Confidence:** 5
**Preliminary Rating:** 3
**Recommendation:** Poster

**Summary:**

The authors present a study on the effect of using additional patient data, such as an indicator of their skin tone, to enhance the performance of the image-only models trained on dermatology images. Their findings suggest that, image only models conditioned using patient information (particularly skin color information) through a multiplication fusion strategy performs substantially better than image only models.

**Strengths:**

- The authors present a simple yet effective set of experiments to  highlight the effectiveness of different metadata fusion approaches in increasing the diagnostic performance of skin image classification models.
- Problem is a hot topic in dermatology

**Weaknesses:**

- Technical novelty is limited as all these fusion models are known to be effective strategies to boost the performance of classification models.
- Design of the cross validation sets can be presented in more detail.
- A key experiment is missing.

**Detailed Comments:**

- Test dataset partitioning strategy may introduce bias if diagnostic label distribution was not balanced between different cross validation sets (e.g. stratified cross validation).
- A key experiment is missing. The authors should have trained independent models for each of the 3 skin tone groups and compare the performance against the fusion models. Since the authors use frozen feature extraction models followed by trainable classification layers, the overhead of having 3 different models will be minimal and this scenario would also be realistic.

**Justification Of The Preliminary Rating:**

The paper presents a technically simple yet useful approach to show how skin tone information can be incorporated to the diagnostic classification problem.
It is necessary to see a few more experiments to understand the effectiveness of the proposed

**Questions To Address In The Rebuttal:**

- Comparisons against 3 independent models for each the 3 FST groups in DDI.

**Special Issue:**

No

---

> ### Author Response · Authors · 2024-03-18
>
> Thank you for your thoughtful review and feedback on our paper. We appreciate the opportunity to address the concerns raised. Please see our comments to the reviewer's feedback as follows:
>
> Technical Novelty:
> We acknowledge the reviewer's observation regarding methodological innovation. Our aim was to apply known effective strategies within a novel context— the use of skin tone information for improving malignancy prediction. We have clarified this aim in the revised manuscript and emphasised the practical significance of our findings in the context of skin cancer detection.
>
> Cross-validation Design Details and Concerns:
> Our dataset was split into training and test sets, with the training set comprising 590 images and the test set 66 images. To further ensure the integrity of our evaluation process, we employed ten-fold cross-validation within the training set. The training set was randomly divided into 10 smaller sets. The process repeated 10 times, with each of the 10 folds serving as the validation set once, and the remaining 9 folds forming the training set. This methodological choice was particularly helpful when we had a small dataset and it allowed for assessment of the model's performance across multiple subsets of the data, enhancing the stability of our findings. For the composition of the test set, we adopted a deliberate and systematic approach to maintain a balanced representation of the three skin tone types, allocating a 1:1:1 ratio (22 images per skin tone type). This strategic allocation was designed to mitigate the risk of bias in model evaluation by ensuring that each skin tone type was equally represented, thereby promoting fairness in the assessment of model performance across the three skin tones.
>
> Experiments on each of the three skin tones:
> Given the constraints of our dataset's size, with 208 cases for light, 241 for light brown, and 207 for dark skin tones, training separate models for each group posed a risk of overfitting, particularly due to the architecture of our model (that consists of both a feature extraction part and a classification part with three layers). To mitigate the risk of overfitting, we explored strategies including training without pre-trained weights, with pre-trained weights, and with frozen weights for the feature extraction component. Training separate models on the relatively small subsets (specific to each skin tone) would not allow the models to generalize effectively, primarily due to the limited diversity and number of samples in each skin tone category.

---

### Official Review · Reviewer_fyya · 2024-03-01

**Confidence:** 4
**Preliminary Rating:** 3
**Final Rating:** 4

**Summary:**

The authors focus on the classification of skin malignancy in the context of the samples depicting skin of different tones. The authors evaluate different methods of fusing the information on skin tone to reduce the heterogeneity of the performances in the skin-tone subgroups. Several architectures are evaluated and the AUC scores per subgroup are reported. The significance test is performed in all the experiments. The reported results show the homogeneity of the results across sub-groups illustrating the interest of the method.

**Strengths:**

The paper is generally clear, the goal is well presented. The authors run a fair amount of experiments, each time with a significance test. It gives a good picture of the performances and the conveyed message.

**Weaknesses:**

There are two major weaknesses. First, the authors propose a method of fusing images with both the skin type and cancer type labels. This raises the question of the contribution of each of the label. That is, it is not clear, whether the skin tone alone allows to achieve the same level of performance if no cancer type is provided and vice versa.

Second, the literature related to skin-tone unfairness is worth more in-depth discussion. As this remains the main target of the paper, it would be useful if other methods and practices could be presented.

**Detailed Comments:**

The authors focus on the skin-tone unfairness in the malignancy classification. It would be useful to provide more information on the state-of-the-art running practices (e.g., 10.1109/NER52421.2023.10123788, 10.1007/978-3-030-59725-2_31).

For the experiments, the authors rely on the dataset from Daneshjou et al. However, little is said about data specifics, leaving the reader to refer to the original paper. It would be helpful if the authors gave more details about the dataset, given that the original dataset contains a different number of images (597 vs 656). How was the training dataset? How many cancers were in the subgroups?

The authors use ten-fold validation which is useful. However, I'd like the authors to comment on the choice of 10 folds (compared to 5) as it does lead to a considerably small test dataset.

The authors propose fusing images with cancer type and skin tone type labels. It would be useful if the authors provided results and discussed how the algorithm behaves with one of the two labels only.

The authors use early stopping. Could the authors provide more details about the number of epochs required? Other implementation details (e.g., learning rate, optimizer) could be helpful.

Finally, proof-reading is advised to prevent typos (e.g., "Proprecessing" in Figure 1), misspellings, typos, and redundancy in citations (e.g., "Ricci Lara et al.(2022) (Ricci Lara et al., 2022).").

**Justification Of Final Rating:**

I thank the authors for providing a feedback on the review.

While there are still some concerns with regard to the contribution of different labels during the training, I believe the paper is a fine conference material.

**Justification Of The Preliminary Rating:**

The paper appears quite concise and clear. However, the main point of homogeneity across different skin-tone types appears to be shadowed to the lack of details in state of the art and the mixture with cancer type in experimental setup.

**Questions To Address In The Rebuttal:**

I would like the authors to address the major weaknesses described above.

**Special Issue:**

No

---

> ### Author Response · Authors · 2024-03-18
>
> Thank you for your thoughtful review and feedback on our paper. We appreciate the opportunity to address the concerns raised. Please see our comments to the reviewer's feedback as follows:
>
> Contribution of the Skin Tone and Cancer Type Labels:
> Thank you for your insightful comments. In the development of our algorithm, our objective was to explore the synergistic potential of combining both skin tone and cancer type labels to improve malignancy classification. However, we acknowledge the importance of understanding how the algorithm performs with independent labels of skin tone and cancer type. Due to the scope of our initial study and the constraints we faced, this comprehensive analysis was beyond our immediate capacity. We plan to undertake this suggested exploration as part of our future work. For the current study, we have included all the detailed results of our experiments in the appendix.
>
> Skin Tone Unfairness in Discussion:
> In the related work section, we have now included the paper by Medero et al. (2023) to contrast their image-only approach with our multi-modal fusion model. While Medero et al. relied solely on image data and employed adversarial training to mitigate bias in skin color recognition, our approach combined metadata with images to provide additional context crucial for accurate classification. Although the work by Kinyanjui et al. (2020) primarily centers on classifying skin tones in the context of skin diseases, we briefly acknowledge their contribution to recognizing the significance of skin tones in dermatological studies. In addition, we have incorporated a table in the appendix summarising the outcomes of our research alongside other studies employing the same DDI dataset, using state-of-the-art methods.
>
> Data Details:
> We have now added a detailed description of the dataset composition, including the count of images used and the distribution of cancer types across skin tone subgroups.
>
> Choice of ten-fold cross-validation:
> In our study, we chose ten-fold cross-validation to ensure the reliability of our findings. This decision was based on its effectiveness in reducing estimate variance, providing a more stable evaluation of the model's predictive performance. We recognise the concerns regarding class imbalance and its potential impact on cross-validation results. However, our dataset exhibits only a moderate level of class imbalance, and thus the risk of skewed performance metrics should be mitigated.
>
> Early Stopping and Implementation Details:
> For a detailed overview of our implementation, please refer to Appendix Table A1. To ensure optimal training for all models, we set an upper limit of 1000 epochs for training. Upon review, we found that the actual stopping point varied across models, ranging from 40 to 100 epochs. this information has now been added in the paper.
>
> Proof-reading:
> We have proofread the paper to correct the typos and redundancies in citations as advised.

---

### Official Review · Reviewer_iqJB · 2024-03-04

**Confidence:** 4
**Preliminary Rating:** 3
**Recommendation:** Poster
**Final Rating:** 4

**Summary:**

The authors have studied the impact of multimodal fusion on model fairness on a public skin cancer dataset. The authors conducted comprehensive experiment to address the research questions on feature fusion strategies and model fairness.

**Strengths:**

The paper is well-structured and written. The experiment setup is well-designed and clearly described. Despite limited methodological innovation, this paper presents insightful findings on the topic of feature fusion and skin cancer detection.

**Weaknesses:**

- Limited methodological innovation.
- The authors did not consider attention mechanisms when evaluating representative feature fusion methods, making the work less thorough.
-  It remains a question whether the findings of this work is generalizable. The findings about the effect of skin tone is only evaluated on one skin cancer dataset. The insights about feature fusion is only assessed in the context of skin cancer.

**Detailed Comments:**

- As the authors mentioned previous research (Hohn et al, 2021) found that fusing age and sex into a model did not perform as well as an image-only model. It is valuable to see if this is the case in this study. This may further contrast the added predictive value of the skin tones.
- I could not find table A1 in the submission somehow.
- On page 3, "These models utilized a common set of hyper-parameters during training to ensure a fair performance assessment". I doubt if using the same hyperparameters during training would ensure fair comparisons as different networks may have different optimal hyperparameters. Would like to hear any thoughts from the authors on this?
- It would be relevant to put the results of this study in context with other state-of-the-art methods using a table.
- Why is test-time augmentation necessary?
- On page 6, for "we trained ..... (3) with frozen weights", do the authors mean "fully frozen"? If so, why would you re-train it as the weights would not be updated anyway?
- In the text, the authors mentioned the highest AUC being 92%, which does not match with the results in Figure 2 (i.e., 89%) if I am not mistaken.
- In the results section, please clarify whether the presented results are from the cross-validation or the test set.
- I found no evidence in the paper supporting the following statement in the conclusion: "This suggests the fixed pre-trained weights are advantageous when data is scarce". I suggest either providing ample facts to underpin the finding or reconsidering the statement.

**Justification Of Final Rating:**

Many thanks to the authors for addressing most of my concerns. Despite limited methodological innovation, this paper could provide valuable insight into feature fusion methods in the context of skin malignancy classification.

**Justification Of The Preliminary Rating:**

Despite limited methodological innovation, this paper presents insightful findings on the topic of feature fusion and skin cancer detection. However, it remains a question of how generalizable these findings are, as they are only evaluated on one dataset. And there are also many things that could be improved as I mentioned in the detailed comments.

**Questions To Address In The Rebuttal:**

see above.

**Special Issue:**

No

---

> ### Author Response · Authors · 2024-03-18
>
> Thank you for your thoughtful review and feedback on our paper. We appreciate the opportunity to address the concerns raised. Please see our comments to the reviewer's feedback as follows:
>
> Methodology Innovation:
> We acknowledge the reviewer's observation regarding methodological innovation. Our primary focus was on exploring the impact of multimodal fusion on model fairness rather than introducing novel methodologies. We have clarified this aim in the revised manuscript and emphasised the practical significance of our findings in the context of skin malignancy detection.
>
> Attention Mechanism:
> Attention mechanisms, while powerful, introduce additional layers of complexity and parameters to the model. Our initial experiments suggested that the incorporation of attention mechanisms would necessitate a significantly broader exploration of model architectures and hyperparameters to optimize performance and fairness effectively. We felt this would exceed the scope of our current research and could potentially obscure the primary focus on basic fusion strategies and their direct impacts on fairness. We understand the importance of attention mechanisms in enhancing model performance. Therefore, we have recommended this area as a future research direction, particularly for studies with access to larger and more diverse datasets where the benefits of attention mechanisms can be more thoroughly evaluated.
>
> Generalizability of Findings:
> We understand the concern regarding the generalizability of our findings. To address this, we have included in the discussion of the revised manuscript that elaborates on the potential applicability of our insights to other datasets and domains, while acknowledging the need for further research to validate these findings across diverse contexts
>
> Previous Research Comparison (Hohn et al., 2021) :
> Thank you for highlighting the importance of comparing our work with Hohn et al. (2021), specifically regarding the integration of age and sex as predictive features. Our study's reliance on a publicly available skin cancer dataset presented a unique constraint: the dataset did not include age and sex information among its shared metadata. This omission limited our ability to directly assess the impact of these demographic features on model fairness and performance in our experiments. Recognising its potential value, we recommend future research in this area, in the discussion section, to use datasets that include a broader range of metadata, including age and sex.
>
> Table A1:
> We apologise for the oversight and have corrected the missing Table A1 in the supplementary materials.
>
> Common set of hyper-parameters:
> Using a uniform set of hyperparameters across models allowed us to establish a consistent baseline for comparison. This approach minimises the variability that could arise from model specific optimisations. thereby providing a clearer view of the intrinsic effects of the fusion strategies on model fairness and performance. However, we recognise that different networks may have different optimal hyperparameters, and have included this as a limitation in our study.
>
> Table for Comparison of Literature:
> We have now incorporated a table in the appendix summarising the outcomes of our research alongside other studies employing the same DDI dataset, using state-of-the-art methods.
>
> Test-Time Augmentation (TTA):
> Using TTA mirrors the data augmentation techniques used during the training phase, ensuring that the model is evaluated under conditions similar to how it was trained. This consistency helps to maintain the model's performance stability by exposing it to a variety of transformations that it has learned to generalise from during training. Hence, it enabled the model to better adapt to and recognise new and unseen variations of image data.
>
> Frozen Weights:
> By frozen weights, we refer to the technique where the parameters of certain layers in the model are kept constant. This approach does not imply that the entire model's weights are frozen, but rather it applied to the layers associated with feature extraction, which have been pre-trained on a large dataset.
>
> AUCs:
> Upon further check, it has been confirmed that the AUC percentages detailed in the text are consistent with those depicted in Figure 2. For a comprehensive view of all AUC scores, please see the tables provided in the appendix Table A5.
>
> Results:
> To address the confusion, we confirm that all results presented in our study, including those mentioned in the text, tables and figures, were derived from the test set. We have now added more detailed information about the dataset in the paper.
>
> Citation:
> We apologise for the oversight regarding the missing citation. The reference has now been added to the paper: "On Pre-Trained Image Features and Synthetic Images for Deep Learning."

---

> > ### Comment · Reviewer_iqJB · 2024-03-26
> >
> > Many thanks to the authors for addressing most of my concerns.

---

### Comment · Area_Chair_wJpy · 2024-03-18
**Follow-up discussion**

Dear reviewers,

The authors prepared a rebuttal based on your comments (but possibly, it was not visible to you until now). We invite you to reply to their responses and change your ratings if necessary. Please do so before March 27th.

---

> ### Author Response · Authors · 2024-03-18
>
> Dear chairs at MIDL,
>
> Sorry for any confusion. I did provide our rebuttal.
> I believe we need to make the comments visible to everyone? We have changed the setting to make our comments visible to everyone.
>
> Yours sincerely,
> Iris

---

> > ### Comment · Area_Chair_wJpy · 2024-03-18
> >
> > Thank you for clearing this up, I am able to see your comments now and I think the reviewers should be able to also. I will update my post.

---

> > > ### Author Response · Authors · 2024-03-28
> > >
> > > Dear chairs at MIDL,
> > >
> > > We greatly appreciate all reviewers' valuable feedback. In response, we have incorporated additional content as suggested. Unfortunately, this resulted in exceeding the 8-page limit. To comply, we've moved some detailed information to the appendix. Our latest version now meets the 8-page requirement. However, currently we are unable to upload our document. Can we ask if it is possible to submit the latest paper to the system again, please? We are very sorry.
> > >
> > >
> > > Kind regards,
> > > Authors

---

> > > > ### Comment · Area_Chair_wJpy · 2024-03-29
> > > >
> > > > Dear authors,
> > > >
> > > > The deadline for revising for papers has passed now so I don't think you can update the paper anymore, furthermore the other reviewers will not be able to review your changes.

---

### Meta-Review · Area_Chair_wJpy · 2024-04-01

**Recommendation:** Accept (Poster)
**Confidence:** 3

**Metareview:**

The paper explores how adding skin tone affects skin cancer classification. The reviewers agreed on the clarity of the paper, and useful insights being presented. Several reviewers mentioned lack of methodological innovation. Some concerns to address were generalizability of the findings, missing baseline experiments to provide insights, and engaging with related literature. After the revision period, some of these concerns were addressed, leading to improved scores for some of the reviewers.

I believe the paper could still be stronger by using more datasets and using relevant baselines. Beyond what the reviewers have discussed, I would also recommend the authors to look into the problem of “shortcuts” which could be influencing the results.  Overall I think the research is still worth discussing at the conference so I would cautiously recommend acceptance for the version present on OpenReview*.

*The authors wanted to do a further update of the paper after the deadline had passed and upload was no longer possible on OpenReview.

---

### Decision · Program_Chairs · 2024-04-06

Accept (Poster)